# Periodontal Behavior and Patient Satisfaction of Anterior Teeth Restored with Single Zirconia Crowns Using a Biologically Oriented Preparation Technique: A 6-Year Prospective Clinical Study

**DOI:** 10.3390/jcm10163482

**Published:** 2021-08-06

**Authors:** Blanca Serra-Pastor, Naia Bustamante-Hernández, Antonio Fons-Font, María Fernanda Solá-Ruíz, Marta Revilla-León, Rubén Agustín-Panadero

**Affiliations:** 1Department of Stomatology, Faculty of Medicine and Dentistry, University of Valencia, 46010 Valencia, Spain; Blanca.Serra@uv.es (B.S.-P.); naibus@alumni.uv.es (N.B.-H.); antonio.fons@uv.es (A.F.-F.); ruben.agustin@uv.es (R.A.-P.); 2College of Dentistry, Texas A&M University, Dallas, TX 75246, USA; revillaleon@tamu.edu

**Keywords:** finish line, anterior crowns, BOPT, periodontal health

## Abstract

Objectives. The aim of this study was to analyze the behavior of the periodontal tissues around teeth in the anterior region when restored with zirconia single crowns, using a biologically oriented preparation technique (BOPT), over a 6-year follow-up. Methods. The study investigated tooth-supported single crowns in the anterior region that were fabricated with a zirconia core and feldspathic ceramic covering, in 34 patients. Follow-up analysis took place annually for 6 years, assessing periodontal responses by evaluating the following variables: plaque index (PI); probing depth (PD); gingival index (GI); gingival thickness adjacent to the restoration; and stability of the gingival margin (MS). Any (biological and mechanical) complications were also recorded, as well as the patients’ satisfaction with the treatment. Results. After 6 years’ follow-up, a low mean plaque index was obtained, probing depth was stable, and gingival thickness and margin stability had increased. Complications (biological and mechanical) did not present a statistically significant incidence and a crown survival rate of 97.2% was achieved. Patients’ satisfaction obtained a mean VAS score of 9.04 under 10. Conclusion. Teeth that are prepared with BOPT in the anterior region present good periodontal behavior around the restored teeth, particularly in terms of the stability of the gingival margin and increased gingival thickness. Single crowns prepared with BOPT obtain an excellent clinical survival rate, as well as a high score in patients’ satisfaction after 6 years.

## 1. Introduction

Prosthetic dental treatments should not limit their objectives to achieving the optimal restoration of function and esthetics but should also include biological responses. In this sense, an ideal prosthodontic treatment must achieve a healthy relationship between the prosthesis and the surrounding periodontal tissues [1,2,3,4,5]. Many factors in the design of a fixed prosthesis influence periodontal health: the prosthesis’s cervical emergence profile, the materials used in fabrication, the finish line created in dental preparation, and especially the location of the prosthetic margin [4,5,6,7,8,9,10,11,12,13,14,15,16,17,18,19].

Finish lines are defined according to their geometry, either as horizontal (non-sliding) or vertical (sliding) [20,21]. The main types of horizontal finish line are with a shoulder at 90° or 120°, a beveled shoulder, a classic chamfer, or a modified chamfer. As for vertical finish lines, the best known is the knife-edge finish line. Another approach to vertical preparation is the biologically oriented preparation technique (BOPT) first introduced by Loi, which consists of vertical tooth milling without a finish line [22,23]. The technique eliminates the tooth’s anatomical crown emergence at the cementoenamel junction (CEJ) to make room for the creation of a new emergence profile by the prosthetic crown. At the same time, rotary curettage of the gingival sulcus is performed [24,25,26,27,28]. The technique was inspired by periodontal prosthetic protocols of the 1980s and 1990s, such as rotary curettage and knife-edge dental preparation [8,14,15]. However, the gingival curettage employed in BOPT differs slightly from the classic periodontal curettage. The rotary curettage now used in periodontics is a treatment aimed at the soft tissue of the periodontal pocket, which aims to eliminate ulcerated tissue or chronic inflammation by means of reshaping the periodontium. In BOPT, for patients presenting good periodontal health, rotary curettage produces complete debridement of the epithelium of the gingival sulcus and the junctional epithelium, which exposes the underlying connective tissue. This provokes bleeding in the area, which will subsequently produce a blood coagulate that fills the area of the supracrestal attached tissue. This is subsequently stabilized by the new prosthetic emergence, whereby the coagulate is replaced in time by mature connective tissue, causing new structuring and attachment [16,24,25,29,30,31,32]. In this way, with the help of the provisional prosthesis, the surrounding soft tissues are modified in shape and position, adapting to the shape of the new prosthetic emergence. This will simulate the anatomical crown of the natural tooth, creating a new cementoenamel line, known as the prosthetic cementoenamel junction (PCEJ) [26]. It should be noted that dental preparation and the exact position of the prosthetic crown are not related. Although the reshaping, of tissues includes connective tissue, the prosthetic crown will be positioned according to the biological position of the cementoenamel junction, 0.5–1 mm from the gingival margin [27,28,29,30].

The second classic concept forming the basis of BOPT is knife-edge dental preparation. This type of dental preparation is indicated for teeth exhibiting periodontal problems, where the finish line must be located at the root, this being a more conservative type of preparation. In BOPT, vertical preparation for the prosthesis is performed by milling the tooth without a finish line, to create a vertical plane between crown and root. On teeth prepared with BOPT, there is no discrepancy between the dental stump and prosthesis, thanks to the effect of the sliding union [33,34]. Incorrect marginal adaptation between the restoration and the prepared tooth seems to be the main reason for increased gingival bleeding and periodontal inflammation, which have generally been more related to prosthetic crowns with horizontal dental finish lines [35,36,37,38]. In addition to the type of preparation, the material used also seems to influence the behavior of the surrounding soft tissues [39,40,41,42].

BOPT has exhibited good periodontal behavior around restored teeth in terms of improved soft tissue marginal stability, better gingival scalloping, and better esthetics [1,23,27,28]; but there are no sufficient long-term studies in the literature to be able to confirm this claim.

To evaluate the result of a fixed dental prosthesis treatment, the clinical complications or the behavior of the periodontal tissues are commonly studied; however, it is of great importance to take into account the subjective perception of the patient [43]. Satisfaction can be defined as the quantification of the opinion of patients with the treatment received [44]. Regarding the BOPT technique, there are no articles that evaluate satisfaction in patients treated using the technique in the long term.

The objective of this prospective clinical study with a 6-year follow-up was to analyze periodontal behavior, clinical complications, and patient’s satisfaction regarding teeth restored with single zirconia full-coverage crowns on teeth prepared with BOPT.

## 2. Material and Methods

This prospective clinical study investigated tooth-supported single crowns in the anterior region, where teeth are prepared with BOPT. The study protocol, which fulfilled the guidelines established in the Declaration of Helsinki for experiments involving human subjects, was approved by the University Ethics Committee for Research Conducted in Humans (Reg. No. H1448361523684); the trial was also registered with ClinicalTrials.gov (Identifier NCT04403230).

All participants were provided with full information about the study protocol, and all provided their informed consent to take part. The patients were treated between January 2013 and January 2014.

The inclusion criteria were as follows: patients who had been treated previously by means of a fixed prosthesis in the anterior region (incisors/canines/premolars) that presented biological (secondary caries, pulpitis), esthetic (recessions, discoloration), or mechanical (fracture) problems, or some other type of complication requiring replacement of the prosthesis; patients aged over 18 years; non-smokers or smoking <10 cigarettes per day; adequate periodontal health (0–3 mm probing depth), or patients presenting managed periodontal disease; patients with normal occlusion and/or managed parafunction.

Exclusion criteria were: the impossibility of placing a fixed prosthesis; patients presenting untreated or unmanaged periodontal disease; unmanaged parafunction (bruxism); or severe systemic disease.

When patients had been selected according to these criteria, BOPT was performed, and the restorations were fabricated. The entire prosthetic procedure was conducted by the same clinician, who is fully trained and experienced in BOPT (R.A.-P.). Before starting treatment, periodontal maintenance was supplied to those patients undergoing periodontal treatment. Patients in good periodontal health underwent conventional ultrasonic oral hygiene maintenance.

During the appointment and prior to dental preparation, alginate impressions (Hydrogum, Zhermack, Badia Polesine, Italy) were taken for measuring the dental arch to be treated and the antagonist arch. In addition, an occlusal register was taken regarding maximum intercuspation, and the dental color was registered for fabricating (in the dental laboratory) a self-curing acrylic resin (Sintodent, Sintodents r.l., Rome, Italy) provisional crown with an approximate thickness of 0.3 mm. Before starting treatment, double probing around each tooth that was in treatment was performed (probing to the base of the gingival sulcus, and to the bone), using a calibrated periodontal probe (PCPUNC156, Hu-Friedy, Chicago, USA) for subsequent periodontal evaluations of the tooth, and to determine the limits of dental preparation and the position of the provisional crown margin. Different periodontal and dental anatomical structures were detected, including the alveolar bone, the emergence of the tooth’s anatomical crown, and the CEJ. Before breaking the periodontal attachment with the probe, the gingival sulcus was probed in order to discount the possibility of any periodontal problem. Then, the previous restoration (Figure 1 and Figure 2) was removed before commencing the BOPT preparation protocol, using the specific burs employed in the technique. Dental preparation was performed following the protocol described by Agustín and Solá [26] (Figure 3).

Firstly, a flame-shaped diamond bur of 1.2 mm diameter and 10 mm length, with 100 µm granulometry (FG863G/012C y FG863M/012C, Sweden Martina, Due Carrere, Padua, Italy) was placed intrasulcularly at an angle of 10–15° to the tooth’s axis, with the tip of the bur placed at a subgingival depth of 1 mm beyond the position of the CEJ, reaching the interface of the attached epithelium and connective tissue. This first step detaches the periodontal soft tissue from the tooth, while the external lateral part of the bur de-epithelializes the free epithelium and attached epithelium, using the tip of the bur to reach the deep area of the attached epithelium (reaching as far as the interface between the attached epithelium and connective tissue in order to detach it). With the internal lateral part of the bur, the first millimeter of the tooth’s anatomical crown emergence is eliminated. It is necessary to detach the whole epithelium in order to produce re-epithelialization and renewed periodontal attachment. All the epithelium attached to the tooth, as far as the connective tissue, was detached to provoke bleeding and the production of blood coagulate that would promote cell differentiation and renewed periodontal attachment [24]. The second step consists of placing the bur parallel to the tooth’s axis (at an angle of 0°) inside the sulcus. This helps to eliminate the preexisting finish line and any convexity or horizontal component of the anatomical crown. When the horizontal component is eliminated, this also removes the conventional means of supporting the previous crown that may have been responsible for producing a gap or a poor fit between dental preparation/finish line and the prosthetic crown. A smooth, vertical, axial plane is obtained, topped by a conical surface that will support the prosthetic crown and adapt telescopically to create a good marginal fit [29]. In the third and last step, the bur is placed slightly convergent toward the incisal edge, to obtain the correct entrance path for the crown. This step will create the milled tooth’s convergence (6°). This process is repeated with burs of finer granulometry, to create a fine surface without any roughness.

When dental preparation has been completed, the provisional crown is adapted to the tooth by rebasing with acrylic resin (Sintodent, Sintodents.r.l, Rome Italy). Once the temporary relining material has set, the area corresponding to the prosthetic emergence is filled with flowable composite (Filtek Supreme Flow, 3M, Minnesota, USA), with the purpose of creating the new prosthetic cementoenamel junction (Figure 4). The provisional restoration is then polished with a special laboratory kit for BOPT (BOPT drills; Sweden & Martina, Due Carrere, Padua, Italy).

Then, the provisional crown was cemented with temporary cement (Temp Bond Clear, Kerr Dental, Orange, CA, USA). The provisional crown margin was placed between 0.5 and 1 mm inside the sulcus and remained in place in the oral cavity for a period of 8–12 weeks until the tissues had completely matured [30] (Figure 5). The provisional crowns were designed with a cervical emergence at 45° to the cervical emergence of the tooth axis. For correct tissue maturation, it is crucial to stabilize the coagulate by means of the provisional crown’s cervical emergence [24].

After the provisional phase, impressions were taken to fabricate the definitive crown. To do this, a two-step ‘putty wash’ technique with silicone was employed (Express Penta Putty and Express Penta Ultra-Light Body; 3M ESPE) with the prior placement of a double gingival retraction cord (Ultrapak, Ultradent, California, USA). The definitive crowns were fabricated with a zirconia core (Lava Frame Zirconia, 3M Espe, Seefeld, Germany) and feldspathic covering (Lava Ceram, 3M Espe) (Figure 6). They were cemented provisionally (Temp bond Clear, Kerr Dental, Orange, CA, USA) for the first two months before definitive cementation with a glass ionomer luting cement (Ketac Cem, 3M Espe).

When cemented definitively, a clinical follow-up protocol was scheduled, with a routine check-up one week after cementation and another occurring six months after cementation, with annual follow-up sessions thereafter for 6 years (periodontal and other data were recorded from the first check-up to the final follow-up 6 years later: T1–T6).

The parameters evaluated were the following: the bacterial plaque index (PI), evaluated by means of the Silness plaque index [45], which assesses the patient rather than the treated tooth; periodontal probing depth (PD) using a millimeter-calibrated periodontal probe (Hu-Friedy PCPUNC156); gingival index (GI) [46]; thickness of the gum around the crown; and marginal stability (MS).

Gingival thickness was measured before starting the treatment around the old restorations, as well as after the treatment around the new ones, as follows. A single point, located 2 mm above the gingival margin in the mid-buccal area of each tooth, was measured using a millimeter-calibrated periodontal probe (Hu-Friedy PCPUNC156) under local anesthetic [27]. In order for this measurement to be standardized, an Essix-type splint was made with a perforation in the measurement area (Figure 7).

Marginal stability was evaluated by measuring the distance in millimeters (mm) from the new prosthetic cementoenamel junction (PCEJ) to the edge of the gingival margin using a periodontal probe. Lastly, statistical analysis was performed to detect any association between changes in gingival thickness and the gingival index, in order to discount the possibility that any increased thickness was related to inflammation.

In addition, biological (pulpitis, secondary caries, fracture of the dental stump, etc.) and mechanical (crown fracture, decementation, etc.) complications were analyzed. Survival/failure rates were calculated according to the complications recorded. Successful treatment was defined as any that did not suffer any complications during the study period. Survival was considered to represent any crown that had remained in place during the study period (including crowns that deteriorated without a need for replacement). A restoration was considered to have failed when the crown needed replacing or the tooth extracting.

Patient satisfaction with the treatment received was established as follows: the patient was asked to give a unique score to the treatment received, taking into account esthetic appearance, comfort (in social relationships, oral hygiene), and function (chewing ability and speech). For this assessment, the patient was shown a visual analog scale (VAS), which is an instrument used to quantify a subjective experience like treatment outcome [47]. In this study, the scale was labeled with 0 (not satisfied with the treatment outcome) at one end, and 10 as the best experience (very satisfied with treatment outcome) on the other end. Patients were instructed to mark the line according to their actual feeling.

Descriptive statistics were calculated for each variable, and a linear regression model was created using generalized estimating equations (GEE), evaluating effects by means of the Wald chi-square test and multiple comparisons with Bonferroni correction, the Brunner-Langer nonparametric test, and a nonparametric ANOVA. Survival was analyzed using the Kaplan-Meier method (CI 95%). All parameters related to periodontal status were represented by scores of 0 to 3 (PI and GI) or 1 to 4 (MI and PD). The significance level applied in the analysis was 5% (α = 0.05).

## 3. Results

The total sample consisted of 74 teeth supporting single crowns in 34 patients (21 women and 13 men, aged between 18 and 65 years) treated with BOPT. Over the 6-year study period, 15 crown samples were lost by the end of follow-up (due to patients failing to attend follow-up appointments).

Plaque indices (PI), recorded in the first and second years, showed a score of 0 in 61.8% of patients, a figure that improved significantly by the third year (81.8%), with subsequent values in the following years of between 74 and 78% presenting a PI of 0. At the end of the 6-year follow-up, the highest PI value obtained was 2 in only two patients (7.1%) (Figure 8). The mean PI values were 0.24 ± 0.56 and 0.59 ± 0.82. It was concluded that the positive changes in PI were significant over the complete follow-up period (*p* = 0.001; Brunner–Langer model ATS test).

The probing depth (PD) results were also favorable as, at the end of the 6-year follow-up, 98.3% of the teeth restored with crowns using BOPT remained stable, presenting PDs of 0–3 mm. Higher PD values of 4–6 mm were found in only four patients. Significant changes in PD were not found over the follow-up period, this indicator of periodontal health remaining stable (*p* = 0.881; Brunner–Langer model ATS test).

Regarding the gingival index (GI) results, after the first year, 89.2% of the sample obtained an index of 0, decreasing to 83.6% of treated teeth in the second year. In the third year, the percentage of teeth giving a value of 0 rose again to 90% and remained more or less stable during subsequent years, presenting percentages of between 87 and 88%. The highest GI value obtained during the entire follow-up period was 2, in a low percentage of teeth (1.7%). Overall, the GI mean values ranged between 0.14 ± 0.42 and 0.24 ± 0.57 (Figure 9). The effect of time was analyzed (GEE model), and it was found that teeth prepared with BOPT did not present significant variations in GI over time; in other words, the GI remained stable (*p* = 0.413).

In the total sample of 74 teeth, pre-treatment gingival thickness was 1.26 ± 0.48 mm, increasing to 1.52 ± 0.51 after the first year (an increase of 20.6%). By the end of the second year, the mean value had increased to 1.68 ± 0.57 mm (an increase of 33.3%). After the second year, it showed very slight increases; the increase in gingival thickness compared with the initial value was 0.26 ± 0.22 mm in the first year, 0.42 ± 0.28 mm in the second year, after which mean values stabilized in subsequent years. Statistical analysis showed that the patients showed a significant increase in gingival thickness over time (*p* < 0.001; Wald chi-square test, GEE model) (Figure 10). The most relevant variation in thickness took place during the first year in comparison with the initial pre-treatment value (*p* < 0.001; Bonferroni test) (Figure 11). After the second year, gingival thickness did not undergo any statistically significant changes (*p* = 1.000).

Lastly, statistical analysis investigated any relationship between gingival thickness and inflammation, in order to discount the possibility that gingival thickness increased due to inflammation. No association was detected between changes in gingival thickness and the GI (*p* = 0.403; GEE model).

The stability of the gingival margins around teeth that were restored with crowns and BOPT was 100% during the first year, decreasing to 97.1% by the end of the third year and remaining stable during the rest of the follow-up period (Figure 12). Only two teeth (2.9% of the total) in two different patients presented gingival recession during the 6-year follow-up; these variations in the gingival margin appeared in both patients after the third year and measured 0.5 and 1 mm respectively, a mean of 0.02 ± 0.13 mm. No statistically significant tendency toward recession was found over the follow-up period, confirming gingival margin stability over time (*p* = 0.231; Brunner-Langer model ATS test).

The biological complications that occurred during the study were one case of pulpitis during the first year, and one extraction due to vertical fracture during the second year; no cases of secondary caries were recorded. As mentioned above, there were two biological complications out of a total of 74 crowns (2.7% of the total). It was concluded that the incidence of biological complications was not statistically significant at any of the evaluation times (T1–T6) (*p* = 0.445; Brunner-Langer model ATS test).

The mechanical complications occurring during the study period were one decementation during the second year, one decementation during the third year, and one complete crown fracture during the fifth year, making a total of three complications out of 74 crowns, 4.1% of the total sample. The incidence of mechanical complications was not found to be statistically significant at any of the evaluation times (T1–T6) (*p* = 0.602; Brunner–Langer model ATS test).

The probability of restoration survival over the 6-year follow-up was 97.2% (Kaplan–Meier method), due to two complications requiring extraction and one crown fracture.

Success was defined as any crown that did not suffer any biological or mechanical complication. In this way, a success rate of 93.1% was obtained, due to the five complications recorded.

Patients’ satisfaction with treatment by means of single crowns, placed on teeth prepared with BOPT, obtained a mean VAS score of 9.04 ± 1.00, and a median score of 9 (half the patients gave scores of 9 or more).

## 4. Discussion

Restoration with full coverage crowns on teeth that were prepared with a finish line has been widely analyzed in terms of the prosthetic crowns’ mechanical behavior [23,24,25,26], but few studies have investigated the behavior of the tissues surrounding the prepared and restored teeth [23]. As an alternative approach, BOPT has generated much interest among dentists, due to the optimal periodontal results it achieves [10,16,27,28]. For this reason, the present study evaluated the clinical behavior of tooth-supported single crowns on teeth prepared with BOPT in the anterior region, with a 6-year follow-up. Another documented variable in the present study is patient satisfaction, which is crucial for evaluating the quality of treatment and clinical results from a subjective point of view [48]. There are many studies affirming that the success of a prosthetic treatment is related to survival, its biocompatibility, and, finally, to patient satisfaction [44,49].

The evaluated variables gave clear indications of teeth treated with BOPT; few clinical studies have looked into these variables in the medium- and long term, in relation to this innovative dental preparation technique.

The study’s 6-year follow-up period was longer than any other work on the subject published to date. The sample size was very similar to studies employing similar methods, such as that of Paniz [28]. The clinical parameters that were evaluated have been investigated in similar articles in the literature, although they were always regarding teeth restored with horizontal finish lines [16,17,18,28,39].

According to the results, the plaque index improved significantly during the follow-up period (*p* = 0.001), which could be due to the fact that, with crowns on teeth that are prepared with BOPT, there is no discrepancy between the dental abutment and prosthesis, due to the effect of the sliding union [33,34]. This concept is based on the idea that two parallel vertical surfaces (axial wall with a knife-edge finish/the crown’s internal surface) can maintain primary contact at any point along their length. For this reason, knife-edge dental preparations are more retentive, as the BOPT crown adapts telescopically to the tooth abutment, achieving a better marginal fit and, as a result, less bacterial filtration and less plaque retention [1,29]. Preparations with a finish line, where the crown is supported by a horizontal shoulder that is not always regular and symmetrical, may cause a poor fit or a gap (Figure 13). According to various authors, incorrect marginal adaptation between the restoration and the prepared tooth is the main reason for increased gingival bleeding and periodontal inflammation [13,17,19,36,38,39].

In the present study, no significant changes in probing depth were observed over the 6-year follow-up, with 98.3% of the teeth presenting the same probing depth (ranging from 0–3 mm) as at the start of treatment. However, several studies of teeth with horizontal finish lines have observed an increase in probing depth during the follow-up period [13,17,40,41].

In the present work, the mean gingival index remained stable during the follow-up, with between 83.6% and 90% of the teeth presenting a GI of 0. These are important findings as, according to the literature, the risk of gingivitis is always slightly greater around prosthetic crowns on horizontal dental finish lines, where GI values are usually seen to increase [35,36,37,38]. It should be noted that several previous studies have reported bleeding on probing [6,28,39], a parameter that the present study omitted since periodontal conditions were evaluated by means of the GI.

Gingival thickness has not been a parameter that is commonly reported in previous articles. Nevertheless, some studies record the patient’s gingival biotype, although changes over time have not been reported [6,16]. In the work by Paniz, neither preparation protocols nor provisionalization were described correctly, and changes in several variables were not monitored over time. In the present study, variations in gingival thickness were recorded, showing statistically significant changes (*p* < 0.001), increasing by an average of 33% during the first two years and stabilizing thereafter. The statistical analysis set out to identify any relationship between gingival thickness and GI, in order to establish whether the increases in gingival thickness that were observed might be due to an increase in inflammation of the tissues, as described in two studies by Paniz [16,28]. However, no statistically significant association between increased gingival thickness and GI was detected.

Regarding changes to the gingival margin, significant stability was observed over the follow-up period. Clinical experience of BOPT has shown that gingival thickness increases together with minimal gingival migration in the medium term [1,16,23,26,27,28,31,32]. But when dental preparations with horizontal finish lines are analyzed, apical migration of the gingival margin is a common response over time [10,12,17,36]. Studies subjected to meta-analysis have shown that 40.7% of samples prepared with horizontal finish lines suffer gingival migration, in comparison with 2.9% of the present work’s sample [10,11,12,17,36].

The incidence of complications, whether biological or mechanical, in the present study was not statistically significant. Crowns on teeth prepared with BOPT presented a high rate of survival. According to the literature, the type of restoration material is a factor that influences biological behavior. In the present work, the crowns were fabricated with a zirconia core and feldspathic ceramic covering. A systematic review found that the behavior of tissues adjacent to all-ceramic crowns was significantly better than with metal-ceramic crowns (Figure 14). Moreover, zirconia restorations require less material thickness, making dental preparation less invasive [20,21,42].

Patient satisfaction was assessed using a VAS; this method is used in numerous studies to evaluate the subjective perception of the patient [44,45]. In the present study, patient satisfaction obtained a mean score of 9, but we were not able to compare this score because it was not evaluated in studies that discussed the BOPT technique.

BOPT has gained popularity in recent years, although the longest clinical observation period for restorations on BOPT teeth published to date was over four years; this is now extended to 6 years in the present study. Further long-term studies are needed to confirm the clinical evolution of this treatment and its stability over time.

## 5. Conclusions

Treatment with single zirconia crowns on teeth prepared with BOPT obtained an accumulated survival rate of 97.2%. Accordingly, the incidence of biological or mechanical complications was not statistically significant and the restorations presented good behavior over the 6-year follow-up.Teeth treated with this technique presented improved plaque indices, stable probing depths, increased gingival thickness, and stable gingival margins over the 6-year follow-up.Treatment with a fixed prosthesis using the BOPT technique has a positive impact on patient satisfaction, especially in cases concerning the re-treatment of old fixed prostheses. This fact is due to the esthetic improvement not only of the restoration but also of the surrounding tissues, as it improves gingival quality by thickening the tissue, thus preventing gingival recession.On the basis of the results obtained, BOPT may be recommended in cases requiring re-treatment with prosthetic crowns.

## Figures and Tables

**Figure 1 jcm-10-03482-f001:**
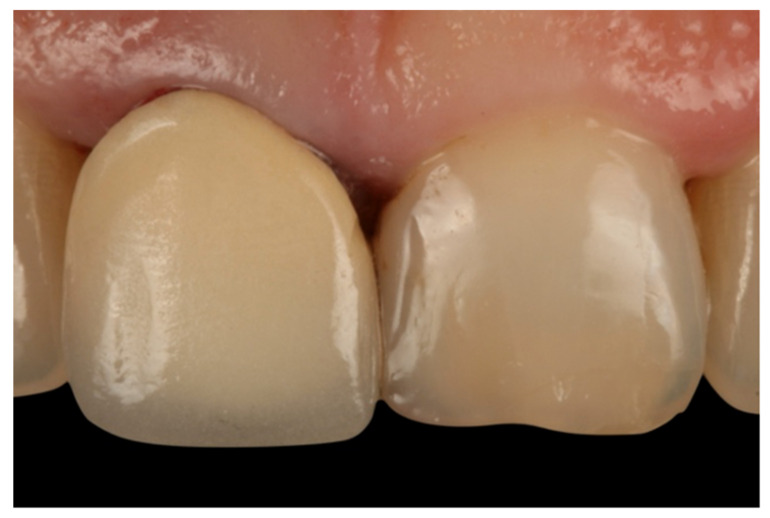
Initial photo of a patient included in the sample, with a full-coverage old restoration.

**Figure 2 jcm-10-03482-f002:**
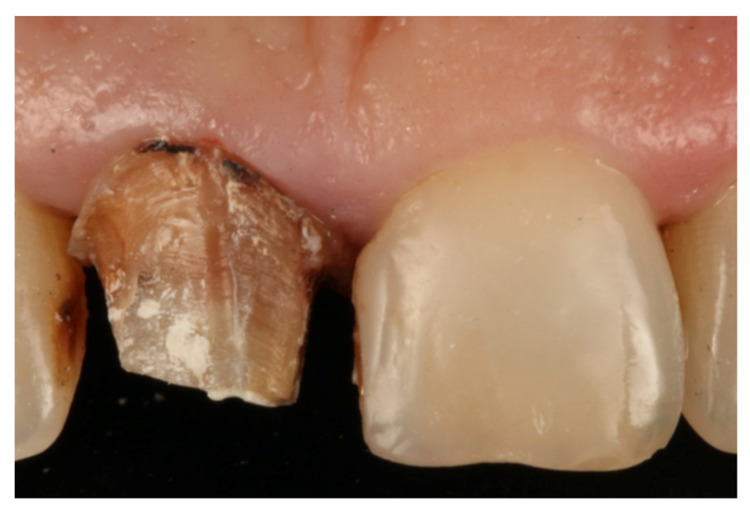
Horizontal finish line after removing the old restoration.

**Figure 3 jcm-10-03482-f003:**
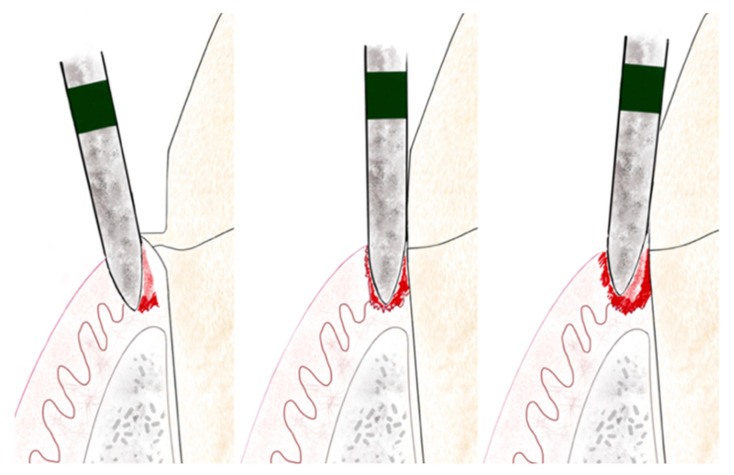
Illustration of BOPT 3-step preparation protocol. The left figure shows the first step, the middle one shows the second step and, on the right, the third step.

**Figure 4 jcm-10-03482-f004:**
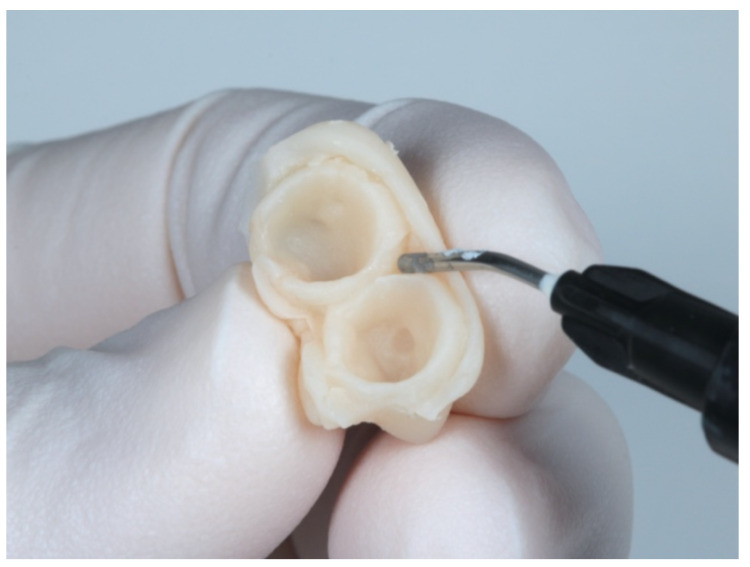
Adaptation of the provisional restoration to create the new emergence profile.

**Figure 5 jcm-10-03482-f005:**
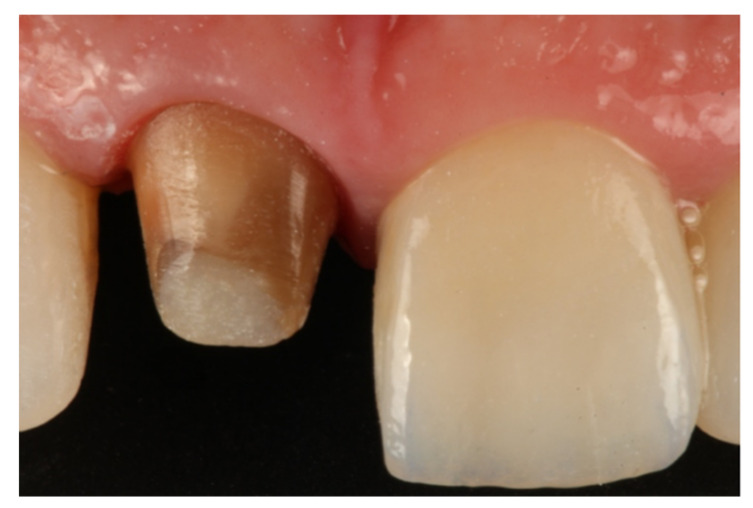
Soft tissue maturation after the provisional BOPT phase.

**Figure 6 jcm-10-03482-f006:**
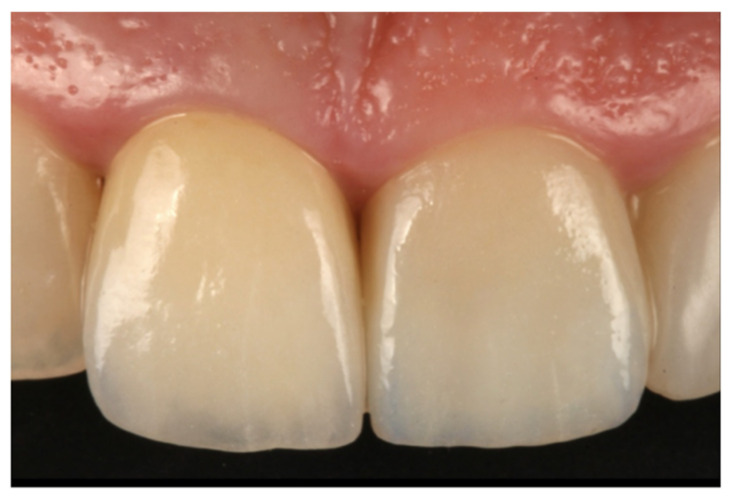
Final zirconia restoration.

**Figure 7 jcm-10-03482-f007:**
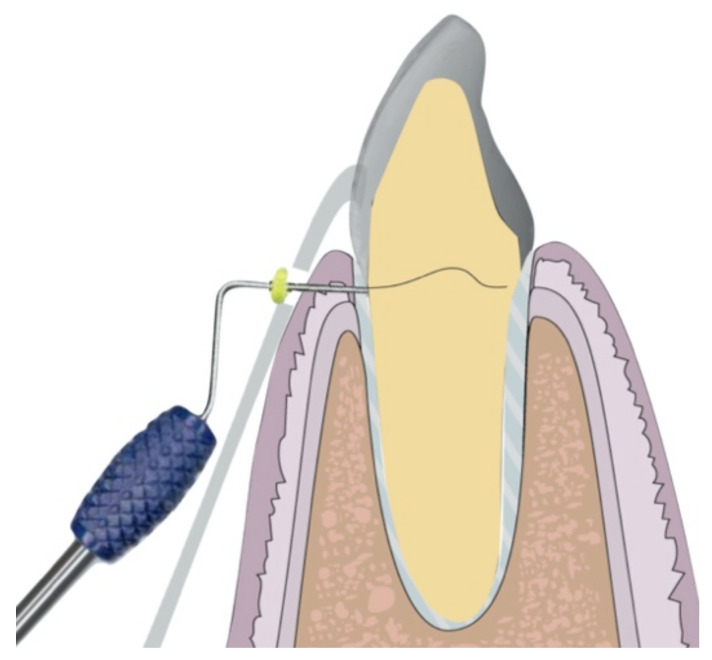
Illustration showing the measurement technique with the Essix-type splint.

**Figure 8 jcm-10-03482-f008:**
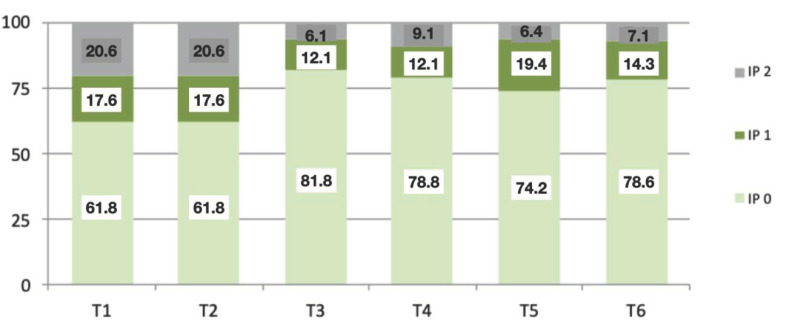
Graphic showing the percentage of patients with different PIs during the follow-up period (from T1 to T6).

**Figure 9 jcm-10-03482-f009:**
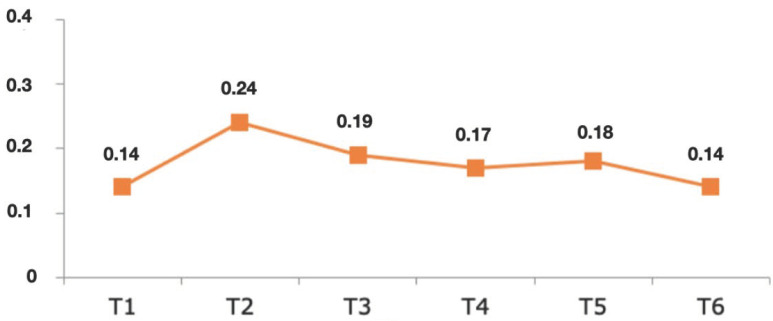
Graphic showing the mean values of GI, from the first year of follow-up (T1) until the sixth (T6).

**Figure 10 jcm-10-03482-f010:**
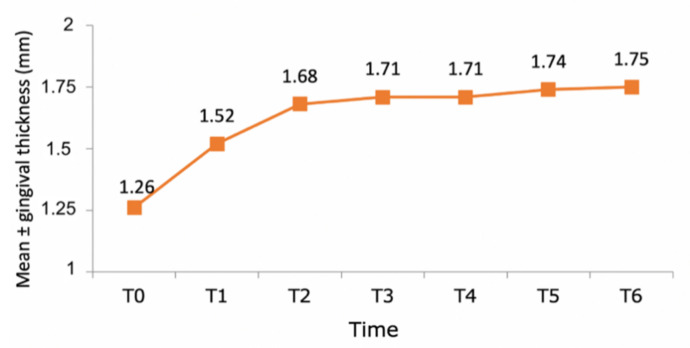
Graph showing the evolution of gingival thickness over a 6-year follow-up (T0–T6).

**Figure 11 jcm-10-03482-f011:**
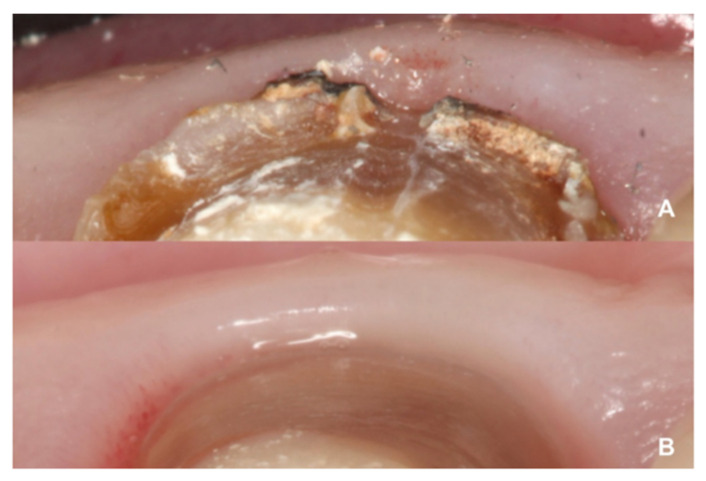
Differences in gingival thickness. (**A**) Tissue thickness adjacent to the horizontal finish line. (**B**) Tissue thickness adjacent to the BOPT.

**Figure 12 jcm-10-03482-f012:**
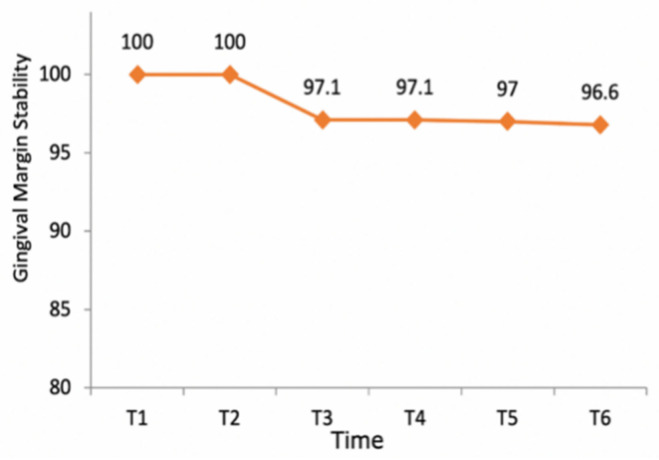
Graph illustrating gingival margin stability over the follow-up period. Note that after the fourth year, stability fell slightly, due to a reduction in the sample size for reasons other than gingival recession (i.e., loss of patients from the follow-up or restoration failure).

**Figure 13 jcm-10-03482-f013:**
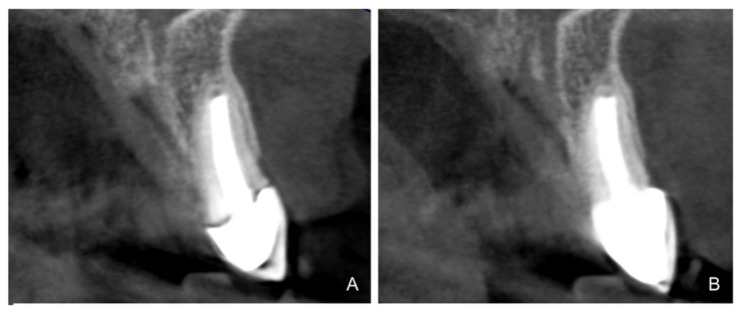
Example before and after re-treatment. (**A**) Initial radiograph of a patient included in the sample with a full-coverage restoration (horizontal line). (**B**) Case after retreatment with a prosthetic zirconia crown, using the BOPT technique. Note the difference in the marginal fit between the two types of dental preparation.

**Figure 14 jcm-10-03482-f014:**
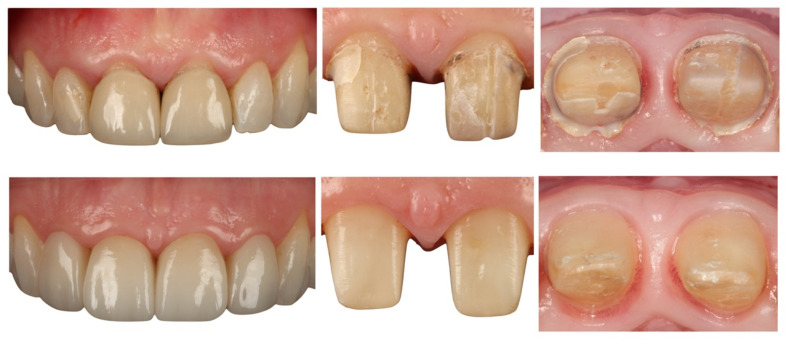
Gingival tissue stability with BOPT treatment.

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
