# Peer review of "Periodontal Behavior and Patient Satisfaction of Anterior Teeth Restored with Single Zirconia Crowns Using a Biologically Oriented Preparation Technique: A 6-Year Prospective Clinical Study"

_jcm, 2021, doi:10.3390/jcm10163482_

Round 1

Reviewer 1 Report

Dear Authors, 

many thanks for submitting your work. I really enjoyed reading it and I believe it represent a very good example of modern prosthetic dentistry.

However I would like to express my points as follow:

  • sample size is relatively low however if so would be important to report a power calculation in order to justify that number. Are there any study with that sample size that can give you significant results? please quote
  • You do not have a control group therefore your findings are essentially related to the single procedure you are describing. Because all your periodontal indices are subjective to the single operator how can you justify that BOPT is for sure a good approach?
  • I would suggest to add one table with all indices recorded per m month over the follow up period. 
  • The provisionals: in the BOPT provisionals restorations are the most important aspect. Would be good for the author to see how you practically adapted the margins of provisional to the soft tissue ( e.g. marginalization)

Kind Regards

Author Response

many thanks for submitting your work. I really enjoyed reading it and I believe it represent a very good example of modern prosthetic dentistry.

Thank you for your words

However I would like to express my points as follow:

  • sample size is relatively low however if so would be important to report a power calculation in order to justify that number. Are there any study with that sample size that can give you significant results? please quote

We are aware that the sample size is not very high, however, it is currently very difficult to find a large sample of patients and that they continue to comply with follow-up appointments for a long period of time. There are similar articles with a slightly larger sample; like that of Paniz; however, this study has a maximum of 1 year of follow-up.

Paniz G, Nart J, Gobbato L, Chierico A, Lops D, Michalakis K. Periodontal response to two different subgingival restorative margin designs: a 12-month randomized clinical trial. Clin Oral Investig 2016; 20: 1243–52.

  • You do not have a control group therefore your findings are essentially related to the single procedure you are describing. Because all your periodontal indices are subjective to the single operator how can you justify that BOPT is for sure a good approach?

The lack of a control group is one of the weak points of our research and that is why in future research we will add it; However, the initial situation of the patients (old fixed prostheses that require re-treatment) is a starting point that has been measured in many of the parameters studied (T0 or pre-treatment measure) and this gives us a comparison of the BOPT technique with the conventional termination line technique (initial situation)

With this study we have been able to observe the stability of the tissues surrounding the restorations (stability of the gingival margin and improvement of gingival health) over time, a very positive data compared to the behavior of fixed prostheses with a conventional finish line.

  • I would suggest to add one table with all indices recorded per m month over the follow up period. 

We added graphics instead of tables because we think that is more instructive and visual for the lecturers; but if needed please let us know.

  • The provisionals: in the BOPT provisionals restorations are the most important aspect. Would be good for the author to see how you practically adapted the margins of provisional to the soft tissue (e.g. marginalization)

We added a figure and some text to explain de adaptation of the provisional.

Reviewer 2 Report

I believe that the study named " Periodontal behavior and patient satisfaction of anterior teeth restored with single zirconia crowns using biologically oriented preparation technique: A 6-year prospective clinical study." will be very useful for dentists.

Author Response

I believe that the study named " Periodontal behavior and patient satisfaction of anterior teeth restored with single zirconia crowns using biologically oriented preparation technique: A 6-year prospective clinical study." will be very useful for dentists.

Thank you for your words; it is a very important work for us.

Reviewer 3 Report

Dear authors,

the present manuscript is a prospective clinical study that aims to analyse the BOPT on anterior teeth with a follow up period of 6 years.

The study is well written and structured but there are some missing points.

Line 15: “Methods….” please add number of patients.

Materials e methods: did you follow a STROBE checklist for prospective observational studies? If yes, please write it in the paragraph and enclosed the PDF of the checklist.

Line 117: alginate, Please add type, brand)

Line 182: please add brand and type of retraction cord

Line 197-199: please explain better the protocol moreover, this could be a bias ad an incorrect evaluation of the gingival thickness due to a not standardization between pre-and post-op. why did you not evaluate the use of a k file or the 3D analysis for the evaluation?

Results: a table could be more useful for the observation. Of data regarding PD, PI,GI during the entire follow up period. Please add a Table.

Figure 7   more useful could be the figures at baseline during the gingival thickness measurement and the follow up with the splint.

Line 396-399 please eliminate Bold

Author Response

Line 15: “Methods….” please add number of patients.

We added it on the manuscript

Materials e methods: did you follow a STROBE checklist for prospective observational studies? If yes, please write it in the paragraph and enclosed the PDF of the checklist.

We are sorry; we didn’t use a STROBE checklist because there are only one group of patients

Line 117: alginate, Please add type, brand)

We added this information on the manuscript (Hydrogum, Zhermack)

Line 182: please add brand and type of retraction cord

We added this information on the manuscript (Ultrapak, Ultradent)

Line 197-199: please explain better the protocol moreover, this could be a bias ad an incorrect evaluation of the gingival thickness due to a not standardization between pre-and post-op. why did you not evaluate the use of a k file or the 3D analysis for the evaluation?

Pre and post-operative measurement technique was the same. Before starting the treatment, the gingival thickness around the old restorations was measured as follows: A single point located 2mm above the gingival margin in the mid-buccal area of each tooth was measured. In order for this measurement to be standardized, an essix-type splint was made with a perforation in the measurement area. K files were not used for the measurement because we thought that the periodontal probe would be firmer and lead to less error. A 3D volumetric analysis was not performed because we still did not have an intraoral scanner at the beginning of the study; but today it would be the technique of choice to record the change in thickness in future investigations.

We explained better the measurement protocol in the manuscript and we also added a figure to better understanding

Results: a table could be more useful for the observation. Of data regarding PD, PI,GI during the entire follow up period. Please add a Table.

 We added graphics instead of tables because we think that is more instructive and visual for the lecturers; but if needed please let us know.

Figure 7   more useful could be the figures at baseline during the gingival thickness measurement and the follow up with the splint.

We conserved the figure 7 but now it is figure 11 in the manuscript because we added more figures to better explain every step.

Line 396-399 please eliminate Bold

We are sorry, we changed it on the manuscript.

Round 2

Reviewer 3 Report

Dear Authors,

congratulations now the article is improved and the description is completed 

Author Response

Thank you very much for your help by reviewing our work, it is very important for us.